# Some Aspects of Development and Histological Structure of the Visual System of Nothobranchius Guentheri

**DOI:** 10.3390/ani11092755

**Published:** 2021-09-21

**Authors:** Dmitry L. Nikiforov-Nikishin, Vladimir A. Irkha, Nikita I. Kochetkov, Tatyana L. Kalita, Alexei L. Nikiforov-Nikishin, Eduard E. Blokhin, Sergei S. Antipov, Dmitry A. Makarenkov, Alexey N. Zhavnerov, Irina A. Glebova, Svetlana V. Smorodinskaya, Sergei N. Chebotarev

**Affiliations:** 1Institute of Biotechnology and Fisheries, Moscow State University of Technologies and Management (FCU), 73, Zemlyanoy Val Str., 109004 Moscow, Russia; niknikdl@rambler.ru (D.L.N.-N.); 750@mail.ru (T.L.K.); 9150699@mail.ru (A.L.N.-N.); a.zhavnerov@mgutm.ru (A.N.Z.); zolotoitina2013@yandex.ru (I.A.G.); kler.smo@gmail.com (S.V.S.); 2Scientific Department, Moscow State University of Technologies and Management (FCU), 73, Zemlyanoy Val Str., 109004 Moscow, Russia; v.irkha@mgutm.ru (V.A.I.); ss.antipov@gmail.com (S.S.A.); 3Federal Research Centre the Southern Scientific Centre of the Russian Academy of Science, Chekhova Ave., 41, 344006 Rostov-on-Don, Russia; holele@mail.ru; 4Department of Biophysics and Biotechnology, Voronezh State University, 1, University Square, 394063 Voronezh, Russia; 5Institute of Chemical Reagents and High Purity Chemical Substances of the National Research Centre “Kurchatov Institute”, Str. Bogorodsky Val, 3, 107076 Moscow, Russia; makarenkovd@mail.ru; 6Management Department, Moscow State University of Technologies and Management (FCU), 73, Zemlyanoy Val Str., 109004 Moscow, Russia; Chebotarev.sergei@gmail.com

**Keywords:** killifish, embryogenesis, evolutionary aspects, morphology, morphometry, lens, elemental composition

## Abstract

**Simple Summary:**

In this work, for the first time, we studied some aspects of the development of the visual system of the annual killifish (*Nothobranchius guentheri*) in the embryonic and postembryonic periods. The morphoanatomical features of the eye during the growth and maturation of the fish are determined. In addition, the normal histological structure of the main elements of the eye is studied. Age-related changes in the elemental composition of the lens of the eye have been studied. The data obtained make it possible to assert the important role of the visual system for the survival of fish in shallow ephemeral pools. In such environmental conditions, the efficacy of the visual system is a factor in survival and evolution.

**Abstract:**

In this, work some aspects of the development of the visual system of *Nothobranchius guentheri* at the main stages of ontogenesis were described for the first time. It was possible to establish that the formation of the visual system occurs similarly to other representatives of the order *Cyprinodontiformes*, but significantly differs in terms of the individual stages of embryogenesis due to the presence of diapause. In the postembryonic period, there is a further increase in the size of the fish’s eyes and head, to the proportions characteristic of adult fish. The histological structure of the eye in adult *N. guentheri* practically does not differ from most teleost fish living in the same environmental conditions. The study of the structure of the retina showed the heterogeneity of the thickness of the temporal and nasal areas, which indicates the predominant role of peripheral vision. Morphoanatomical measurements of the body and eyes of *N. guentheri* showed that their correlation was conservative. This indicates an important role of the visual system for the survival of fish in natural conditions, both for the young and adults. In individuals of the older age group, a decrease in the amount of sodium (Na) and an increase in magnesium (Mg) and calcium (Ca) were found in the eye lens. Such changes in the elemental composition of the lens can be a sign of the initial stage of cataractogenesis and disturbances in the metabolism of lens fibers as a result of aging. This allows us to propose *N. guentheri* as a model for studying the structure, formation, and aging of the visual and nervous systems.

## 1. Introduction

The visual system of vertebrates is one of the most conservative, since the vitality of the organism and its interaction with the environment, such as reproduction, migration, search for food, and physical activity, depend on its work [1]. Elements of the visual system of fish are laid down at an early stage of embryogenesis. The quality of fish vision depends on many factors, such as the density of the retinal photoreceptors, types of photoreceptor cells, the size of the eye, the lens, and the vitreous body [2,3]. It should be noted that the eye is not the main sense organ for all groups of fish, nor for this class of vertebrates as a whole; thus, fish have wide variations in eye size and shape.

The *Nothobranchiidae* family of fish is a large group, mainly living in the shallow ephemeral pools of northern Africa, in connection with which they have formed a unique mechanism of reproduction and development, where the individual stages of ontogenesis are close to each other [4]. Representatives of this family are a convenient object for studying the processes of evolution, development, and aging [5,6]. Members of the genus *Nothobranchius* are characterized by a wide variety of karyotypes, which may play an important role in the adaptation of fish to unstable environmental conditions [7]. Their short life cycle is due to living in dry bodies of water, which are only filled with water for a certain period of the year. The food supply for such water bodies is not significant and consists of insects and their larvae [8]. Under such conditions, the efficacy of the visual system is a factor in survival and evolution [9]. In shallow ephemeral pools, the amount of solar insolation and the amount of incoming ultraviolet (UV) rays can be destructive if the fish has not formed an effective defense mechanism [10]. *N. guentheri* have a wide variety of behavioral responses, both in the natural environment and in laboratory conditions. Ethological tests based on the use of different variants of colors as an influencing factor show high learning ability and confirm a high degree of development of the visual system and the corresponding parts of the brain [11]. The intrapopulation hierarchy of *N. guentheri* males is based on the size and color intensity of individual fish, which is also largely determined by the development of color vision [4].

In many vertebrates, long-term exposure to UV rays contributes to the development of lens cataracts [12]. Early stages of lesions affecting lens transparency may be evaluated by the elemental composition, in particular the change in the ratio of potassium (K), sodium (Na), and calcium (Ca) [13,14]. Ultraviolet radiation also leads to a disruption in the structure of lens proteins, e.g., crystallins (most common α-, β- and γ-crystallin) [15]. Preparation of more detailed information about irregularities in the structure and synthesis of proteins may be established through transcriptome analysis, which is currently being studied in the brain [16]. Changes in fish habitat and the pollution of water bodies by technogenic discharges can also be assessed by the chemical composition of organs with slow metabolism, otoliths, and the eye lens [17,18,19,20].

The embryonic development of *N. guentheri* takes a long time (at least 21 days under laboratory conditions) [21], and usually includes three diapauses necessary for the full formation of the larva [22,23]. Such a complex development mechanism is due to the drying out of temporary water bodies and makes it possible for the fish population to survive long-term unfavorable conditions without substantial energy losses [4,9]. The last diapause preceding hatching is the most stable and can last more than six months [18].

The visual system of *N. guentheri* has great importance to its lifestyle, as indicated primarily by the size of the eyes in relation to the body size. According to our observations, the eye size of *N. guentheri* changes significantly during life. In many species of teleost, there is a correlation between habitat conditions and morphological characteristics of the eye [24]. One of the characteristic features of the visual system of the Nothobranchius is the significant size of the lens.

The aim of this work is to identify some aspects of the morphoanatomical parameters of the eye of *N. guentheri* and its parts throughout life. The short life cycle makes it possible to study the development of the visual system, starting from the early stages of embryogenesis and ending with the death of fish from natural causes, which, under laboratory conditions, occurs at 10–12 months post-hatching. In addition, the study investigated the content of individual elements in the lens of fish in order to identify the patterns of their exchange and accumulation in different age groups.

## 2. Materials and Methods

### 2.1. Object of Study

The study complied with the guidelines of the Local Ethics Commission of the Institutional Review Board of Moscow State University of Technology and Management (approval number 7, 2 February 2021).

The isolate *N. guentheri* Zanzibar TAN 14-02 was obtained from the collection of the Engelhardt Institute of Molecular Biology of Russian Academy of Sciences at the age of two to eight months. This served as the main base of the experiments in 250 individuals. Fish were divided according to sex and size parameters, and were kept in aquariums (45 × 45 × 15 cm, W × L × H) with a volume of 50 L, with 10 fish each of the same size and gender, and constant aeration at a temperature of 22 ± 2 °C and pH 7.2–7.6. The water was changed according to the following scheme: partial, every two days; full, every seven days. The fish were fed live food (*Artemia salina*, *Daphnia magna*, and larvae of the *Chironomidae* family) twice a day, at 12:00 and 18:00.

### 2.2. Embryology

Eggs for the study of embryogenesis and the development of the visual system were obtained from the group of *N. guentheri*, consisting of one male and two or three females, 5–7 months old, more than 3 cm in size. Individuals were placed in aquariums with a volume of 50 L. The water layer did not exceed 10–15 cm, at the bottom of which there was a container with a peat substrate crushed to 0.5–0.8 mm. The eggs were taken every seven days and incubated in a moist peat substrate (60–70% water by weight) at a temperature of 24–25 °C.

Control of embryotic development was performed every six hours during the first three days (before the beginning of diapause I) and on days 9–14 (from the beginning of diapause II to the beginning of III). The remainder of the time, control was carried out in races once per day.

*N. guentheri* was grown according to the method of Genade et al. [23]. On days 21–24, embryos at stage III of diapause were selected. Then, they were transferred to incubators with distilled water for hatching.

### 2.3. Histology

To obtain histological sections seven-month-old males without visible damage were selected. The fish were sacrificed in MS-222 solution (250 mg/L), after which they were fixed in 4% neutral formalin solution for 24 h at room temperature. Then, tissue samples were dehydrated in a series of graduated alcohols and embedded in paraffin. To obtain the eye in the horizontal plane [25], serial sections of the frontal plane (anterio-posterior plane) of fish (4 μm) were stained with hematoxylin and eosin (H&E) and examined under a light microscope. Preparation and staining of histological slides were performed according to Suvarna et al. [26].

### 2.4. Microscopy

An Olympus BX53 light microscope (Olympus Corporation, Japan, Tokyo) with a Carl Zeiss ERc 5s eyepiece attachment (Zeiss, Germany, Oberkochen) and ZEN lite software (Zeiss, Germany) was used for microscopy of eggs and histological preparations. Egg development was examined in plates with 96 wells.

In total, 10 sections (*n* = 10) obtained from 5 fish were examined. The morphometric measurement of the layer thickness in the temporal (caudal) and nasal (cranial) zones of the retina was performed on 3 sections (*n* = 3) (50 measurements per section).

### 2.5. Fish Photography and Morphometric Measurements

To assess the morphoanatomical changes in the fish eye during ontogenesis, individuals of different sizes and age composition were photographed using the techniques described by other authors with some modifications [27,28,29]. The fish were sedated in a solution of MS-222 (0.1 mg/L) [30], after which their physical activity significantly decreased. This made it possible to take photographs without hindrance. The location of the source of artificial lighting was carried out in such a way that the resulting image on the camera was clear and there were no various negative effects (glare, overexposure, etc.). In this work, we used a studio diode ring illuminator Raylab RL-0518 Kit (Russia) with a color temperature of 4200 K (neutral color range) and a Nikon D5000 camera (Japan). For a reliable assessment, the survey was carried out under the same lighting conditions (4200 K) and camera settings (ISO 400–500, F 5.6, 1/60 s) for all the studied fish.

A total of 107 fish were photographed during the study, from which 60 fish were randomly sampled.

Morphometric measurements were performed using ImageJ software (Wayne Rasband (NIH); https://imagej.nih.gov/ij/; access date: 6 June 2021 ). On all the selected photographs of the fish, the parameters shown in Figure 1 were measured and selected based on the described method [31].

### 2.6. Eye Elemental Composition

The elemental composition of the lens was measured in 4 males of each age group (2 and 7 month) (*n* = 4 × 2) by X-ray energy dispersive microanalysis on a two-beam scanning electron microscope (Zeiss CrossBeam 340 with a Schottky cathode) using an X-Max 80 Oxford Instruments detector. To ensure the reliability of the results obtained and to reduce the measurement error, a preliminary calibration of the detector was carried out on standard samples at reduced accelerating voltages of 7.5 and 10 kV. In addition, in the course of measurements, a 25 nm thick carbon film deposited on the surface was taken into account, the signal introduced by the film atoms was subtracted from the total sample signal by specialized software.

### 2.7. Statistics

Comparative analysis of various parameters under study was performed using Student’s *t*-test; the value of *p* < 0.05 was taken as a significant difference. The Shapiro–Wilk test was used to determine the normal distribution of the data.

The correlation between different morphometric parameters and different elements was determined using Pearson’s correlation with Student’s t-distribution to calculate the significance. Paired linear regression was performed for parameters with a significant value of the correlation coefficient. Statistical data processing was performed using GraphPad Prism version 8.0 software (GraphPad, San Diego, CA, USA).

## 3. Results

### 3.1. Embryogenesis

The formation of the visual system of *N. guentheri* begins with the formation of the optic (eye) vesicle at 16–18 h post-fertilization (hpf) (not shown). Further differentiation and formation of the optic vesicle ends at 24–26 hpf (Figure 2a,b). At this stage of development, the anterior part of the brain is clearly distinguishable, which indicates the parallel formation of the visual and nervous systems. At this time, the lens is formed by invagination of ectoderm into the optical cup. Further development is suspended due to the onset of diapause I (72–96 hpf), the duration of which depends primarily on the water temperature.

After the end of diapause, development is accelerated. In the period from 96–120 hpf, further formation of the nervous system occurs, and the front and middle parts of the brain are already distinguishable. In the lens, the process of forming the lens fibers from the cubic epithelium continues. There is a differentiation of the periocular mesenchyme, which includes melanocytes (Figure 2c,d).

Under non-optimal conditions for the course of embryogenesis at this time (five to seven days post-fertilization (dpf)), diapause II occurs. Its duration depends on a large number of factors [32], most of which are insufficiently studied. For these reasons, under laboratory conditions, diapause II can last from several hours to tens of days. In the conditions of our experiment, its completion was noted at 8–12 dpf. After diapause, there is a notable acceleration of organogenesis. By the time of the onset of diapause III (12–14 dpf) in the embryo, the visual system can be considered fully formed, and the main structures of the eye, including the sclera, cornea, and lens, are clearly distinguishable (Figure 2e). Melanocytes, which are part of the choroid, are visibly clearly distributed in connective tissue differentiated from the peculiar mesenchyme. Complete differentiation of all elements of the eye is completed by the end of diapause III (21–26 dpf), which can be determined by the presence of body and head pigmentation in the embryo, which is characteristic of postembryonic fish. In the eye of the embryo, the pupil and the iris are clearly distinguishable (Figure 2f).

### 3.2. Post-Hatching Development

*N. guentheri* larva, during the transition to a free lifestyle, is characterized by positive phototaxis and the dorsal part of the body is covered with melanocytes to protect it from UV radiation (Figure 3a,b). The eye occupies most of the head section of the fish. In our laboratory, 12–24 h after hatching, *N. guentheri* larvae switched to active feeding on Artemia nauplii. At 7–10 days post hatching (dph), the larva acquires some of the features characteristic of adult fish: the formation of a scaly cover, a change in the shape of the mouth, and the shifting of the eye to the ventral side (Figure 3c,d).

### 3.3. Histology Assay

In total eyes, the *N. guentheri* slice shows that the basic structure of the visual system has a standard structure for teleost. Nevertheless, it was possible to identify some features, most of which are associated with the structure of the retina. The lens (Figure 4b,c) has no pronounced features. The lens has clearly distinguishable lens fibers, formed from a single layer of epithelium, which are located at the anterior pole of the eye. The cornea usually consists of three layers: squamous corneal epithelium, corneal stroma, thin corneal. *N. guentheri* has only one well-defined structure of cornea—the corneal stroma poorly differentiated into separate layers. The ligamentum pectinatum was identified at the junction of the iris and cornea (Figure 4f).

The *N. guentheri* retina has a clear-layered structure, including seven layers: ganglion cell layer, inner plexiform layer, inner nuclear layer, outer plexiform layer, outer nuclear layer, inner segment/outer segment of the photoreceptor cells, retinal pigment epithelium/cells (Figure 4d,e). At the same time, the outer nuclear layer (ONL) presents differentiated photoreceptors with well-developed outer segments (Figure 4g), which makes it possible to assert the high quality of vision of this fish species.

Comparison of the temporal and nasal areas of the retina made it possible to reveal a significantly (*p* < 0.05) greater thickness of retina layers in the temporal area (Figure 4l,m). In addition, the retina of the eye has its own choroid rete, similar to *Danio rerio* [33]. Choroid in adult fish restricts the choroid rete from the caudal side and consists of connective tissue and melanocytes. The ciliary marginal zone (CMZ) is observed in the most-peripheral retina (Figure 4h,i).

### 3.4. Morphoanatomical Measurements

Measurement of the features of the eye size to the size of head and body in different-sized individuals of *N. guentheri* males made it possible to establish the conservatism of these parameters.

There was a significant correlation (*p* < 0.05) (Figure 5d) between the morphometric parameters of the body (SI, ICA) and eyes (dO, sdO). The highest values of the coefficient of determination were obtained by comparing the parameters dO and sdO of the eye and head length, 0.6995 and 0.8072, respectively (Figure 5b,c). The coefficient of determination between the parameters of the eye and the standard body length were slightly lower (Figure 5e,f).

### 3.5. Elemental Composition of the Lens

As a result of measuring the elemental composition of fish lens at the ages of two and seven months, it was revealed that some of the elements retain their concentration throughout the life of *N. guentheri*. Such elements include silicon (Si) and sulfur (S). The other elements included in the lens during ontogeny are more volatile (Table 1, Appendix A).

In 2-month old males the detected elements (excluding carbon and oxygen) constitute 2.64% of the weight of the lens composition, while in 7-month-olds, this percentage is 3.2% (Figure 5a,c). Element concentration data can be arranged in the following sequence:For 2-month old: S > Al > Mg > Na > P > Ni > Cl > Ca > Si;For 7-month old: Al > S > Na > Ca > Mg > Ni > P > Cl > Si.

The evaluation of the correlation between all detected elements made it possible to establish the presence of positive significant correlations between the content of the following elements: Ca, Na, and Si (*p* < 0.0001 and *p* < 0.05, respectively); Cl, Si, and P (*p* < 0.05 and *p* < 0.001); S and Si (*p* < 0.01) (Figure 6b and Appendix A).

## 4. Discussion

### 4.1. Embryology and Post-Hatching

The results of this study suggest that *N. guentheri* has an effective visual system, adapted to exist in small ephemeral ponds and participate in the formation of the social hierarchy of males of this species of fish.

At this time, the embryogenesis of *N. guentheri* and the factors affecting long-term diapause have been widely described [8,33,34,35,36]. However, these works do not devote much attention to the development of the visual system.

The data obtained in the course of this work suggest that the process of the embryonic development of the visual system of *N. guentheri* forms similarly to that in all other bony fish species [37], but takes much longer due to the presence of the diapause. The obtained terms of development are typical for the conditions of our laboratory and have been tested many times, but it should be recognized that, in other conditions, the stage of development can vary significantly depending on the chemical and temperature parameters of the environment. Due to the shortened life cycle, by 7 dpf the larva acquires some features typical of juvenile fish. Fish in the juvenile stage can occupy all possible ecological niches of shallow ephemeral pools [24].

The study of the development of the visual system in other members of the genus Nothobranchius shows that differentiation of retina cells does not stop after hatching and continues throughout the life of the fish [38,39]. The emergence of new cells in the fish retina occurs from the proliferation of multipotent progenitor cells located in the CMZ [40,41] and from dividing Müller glial cells [41]. For *N. guentheri*, we have shown the possible presence of this zone, but this fact requires additional research.

The shortened developmental period and the rapid transition to the reproductive stage make it possible to classify this fish species as precocial. The high development rate is an adaptation to life in drying up water bodies [42], where reproductive success directly depends on growth rate and social hierarchy. Based on this, it can be assumed that the visual system of *N. guentheri* is functional at the time of hatching, since it plays an important role in survival in the natural environment.

The closest analogue of non-annual fish may be *Oryzias latipes*, whose stages of development of visual system are completely repeated but differs in terms of individual stages and the entire embryogenesis as a whole [43]. From the group of annual killifish, similar development times are observed in *Aphyosemion gardneri* [44]. The formation of the visual system of *N. guentheri* proceeds in parallel with the formation of the fish brain and, by the time of the onset of III diapause, it is fully functional (phototaxis) [39]. This is typical for most annual fish [24,44].

By the time III diapause begins, the pigment part of the choroid covers most of the eyeball. It has been reported [44] that melanocytes in the choroid prevent light scattering, providing a clearer image. Freshwater ecosystems are characterized by a high degree of UV absorption due to the low proportion of transmitted sunlight caused by large amounts of organic and inorganic particles [45]. However, for the shallow ephemeral pools in which killifish live, the amount of absorbed radiation can be much higher, leading to the formation of a special mechanism that protects the photosensitive cells of the retina primordium. It should be noted that the bottom fish, such as *Scophthalmus maximus*, have a poorly developed pigment layer of the choroid [46]. It is likely that the visual system is also one of the components of the mechanism of hatching the larvae of annual fish, since even under suitable hydrochemical conditions not all fry hatch simultaneously [47]. In this case, suitable illumination conditions may serve as a stimulus for hatching.

### 4.2. Histological Assay

The study of the eye histology of adult *N. guentheri* (seven months) showed that the main elements of the visual system are similar to other fish species [48,49,50]. As noted by Seritracul et al. [2], the structure of the retina is very similar in all vertebrates, including humans. In addition, fish retina has the properties of postnatal neurogenesis and regeneration (due to the presence of stem cells). These facts make the fish retina an excellent model in biomedicine. The detected choroid rete restricts the blood vessels that do not penetrate the retina. This indicates the absence of retina vascularization in *N. guentheri*, as in most other teleost fishes [51,52].

In all studied fish, no visible pathological changes in the structure of eye tissues were observed. Despite accelerated aging and the appearance of age-dependent changes in other fish organs [53,54], the visual system continued to function normally. The reason for this is likely the significant role of the visual system for the survival of the organism in the natural environment [55]. While thinning of the retina layers and a decrease in the pigment epithelium layer are often noted in aged individuals of *Oryzias latipes* and *Danio rerio* [50,56], these changes will occur in a very short period of time preceding the death of a *N. guentheri* due to its short life cycle.

The thickness of the retina and its layers in *N. guentheri* was found to be similar to *D. rerio* (~109 and 113 µm, respectively) [51]. Differences in the thickness of the retina layers in the temporal and nasal areas of the eye are possibly associated with peripheral vision, which plays an important role in various behavioral acts, such as reactions to a predator, and sexual and feeding behavior [8], which is typical for most pelagic fish [57,58]. In particular, the harem sexual behavior characteristic of *N. guentheri* implies the protection of a certain spawning area from competing males, which determines the success of reproduction with the maximum number of females. Under these conditions, vision plays an important role in maintaining social hierarchy. The increased thickness of the cranial part of the retina is most likely a consequence of the ecological adaptability of fish to life in shallow ephemeral pools [59]. In addition, the amount of light is an important factor in the evolution of the eye of all vertebrates [60]. Thus, the development in the conditions of shallow ephemeral pools contributed to the adaptation of the killifish visual system to the conditions of high light insolation. To obtain more complete data on the thickness of the retina layers, additional studies are needed to compare the dorsal and ventral sides.

The similarity of the retina and lens structures between *N. guentheri* and other fish species such as *Oryzias latipes* and *Danio rerio* [43,45] suggests the possibility of using it as a model for studying the structure, development, and occurrence of age-dependent pathologies.

### 4.3. Morphoanatomical Characteristic

Revealing the regularities of the eye size from other morphoanatomical parameters confirms the assumption about the conservatism of the visual system of *N. guentheri*. The ratio of the eye and head parameters does not change during the entire life span, although the fish of order Cyprinodontiformes are characterized by high population variability in a number of other parameters [61]. The relationship between the diameter and area of the eye and the size of the whole body is not so significant due to the variance in body size. All this speaks in favor of the presence of ontogenetic allometry between the dimensional characteristics of the eye and the body of the fish. Allometry in fish has previously been found between body and lens size [62]. It should also be noted that the size of fish eyes shows the greatest variability among other groups of vertebrates [63]. The location and size of the eyes of *N. guentheri* are characteristic of fish feeding on plankton and small benthic organisms.

### 4.4. Elemental Composition

The study of the microelement composition of the lens, as an inert structure of the body, is an effective way to assess age-related changes [19,20]. Since the metabolism of the lens is slow, the substances that enter it persist for a long time, and in many cases, throughout the life of the fish [62]. The elemental composition of inert tissues may reflect the environmental conditions in which fish live, as has been demonstrated for some fish species [19,44]. In addition, pathological changes will occur when the protein composition of the lens is disturbed, which in turn can lead to a violation of transparency [12,64]. Such changes most often occur at the last stages of ontogenesis and can be caused by two main reasons: (i) impaired differentiation of lens fibers from the epithelium in the equatorial zone of the lens (impaired folding and post-translational protein modifications) due to damage, diseases, etc. [65], and (ii) toxic effects associated with the accumulation of pollutants [66,67,68], also leading to disruption of the synthesis of crystallins [69].

According to the results of this work, aluminum, which accumulated in significant amounts in the lenses of seven-month old fish, may be one toxic element that can lead to the formation of cataracts in *N. guentheri*,. The studies carried out made it possible to compile a series of percentage concentrations of elements, which showed that with age, in *N. guentheri*, the concentration of Na decreases, while the concentrations of Mg and Ca increase. Since these elements have an important biogenic role, it is likely that we can talk about a decrease in or impairment of the metabolism of the lens cortex with the age of the fish. It has been reported that calcium accumulation is a sign of the development of the initial stage of cataractogenesis [70]. The non-biogenic elements of the lens also change with the growth and aging of the fish [19]. The accumulation of these elements is primarily associated with the quality of the aquatic environment, as well as with the high permeability of the lens for ions, water, and other small molecules due to ion pumps and gap contacts [64,69,71]. For example, in fish living in anthropogenically polluted waters, there is an increase in the content of heavy metals and other non-biogenic elements in the lens and otoliths [72].

## 5. Conclusions

As a result of this work, the development of the visual system of *N. guentheri* at the main stages of ontogenesis was described for the first time. It was possible to establish that the formation of the visual system occurs similarly to other representatives of this order, but significantly differs in terms of the individual stages due to the presence of diapause. The beginning of the formation of the visual system occurs at 16–18 hpf and the main structures of the eye are formed in parallel with the processes of neurogenesis. The main structures of the eye can be considered formed by the time of the onset of III diapause (12–14 dpf).

The histological structure of the eye in adult *N. guentheri* practically does not differ from most bony fishes, which live in similar environments. The study of the structure of the retina showed the heterogeneity of the thickness of the temporal and nasal areas. None of the studied histological preparations revealed age-dependent changes in the structures of the eye.

Morphoanatomical measurements of the body and eye parameters of *N. guentheri* showed a conservative correlation of these parameters. This indicates the important role of the visual system in the survival of fish in natural conditions.

The elemental composition of the lens changes as the fish grows and matures. In individuals in the older age group, a decrease in the content of Na and an increase in Mg and Ca were discovered. Such changes in the lens can be a sign of the initial stage of the impaired metabolism of lens fibers. The accumulation of aluminum with age is most likely an example of the deposition of abiogenic elements in inert structures of the body.

The study of the mechanisms of the formation of the visual system of *N. guentheri* allows us to assert the great role of vision for the survival of fish in their natural habitat of shallow ephemeral pools, in its participation in sexual selection, and in the social structure of the population. In laboratory conditions, it is possible to study many behavioral reactions in *N. guentheri* in which vision plays an important role. All of this allows us to propose *N. guentheri* as a model for studying the structure, formation, and stages of the onset of age-related changes in the visual and nervous systems.

## Figures and Tables

**Figure 1 animals-11-02755-f001:**
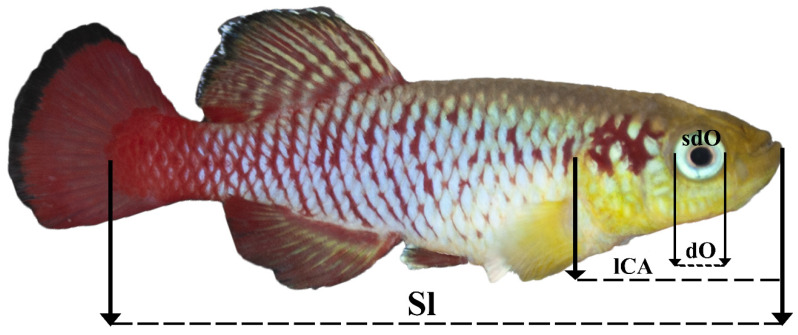
Morphological characteristics measured in this work. Sl—standard length; lCA—head length; dO—eye diameter; sdO—eye area.

**Figure 2 animals-11-02755-f002:**
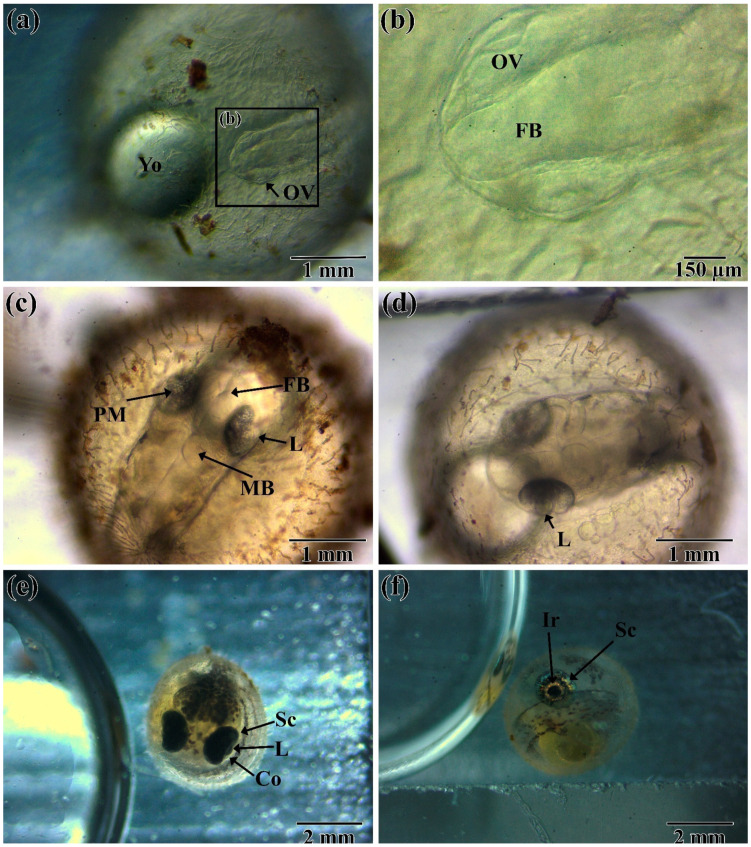
Embryonic development of the visual system of *N. guentheri* in the period from 26 hpf to 21 dpf: (**a**,**b**) 26 to 28 hpf, early neurula, ocular vesicle formation; (**c**,**d**) 96 hpf to 7 dpf early somitogenesis (6–12 somites); (**e**) 10 to 12 dpf retina pigmentation (30–35 somites); (**f**) 21-n dpf stage before hatching (duration depends on the course of diapause). Yo—yolk; Ov—optic (eye) vesicle; FB—forebrain; L—lens; MB—mid-brain; PM—Periocular Mesenchyme; SC—sclera; CO—cornea; IR—iris.

**Figure 3 animals-11-02755-f003:**
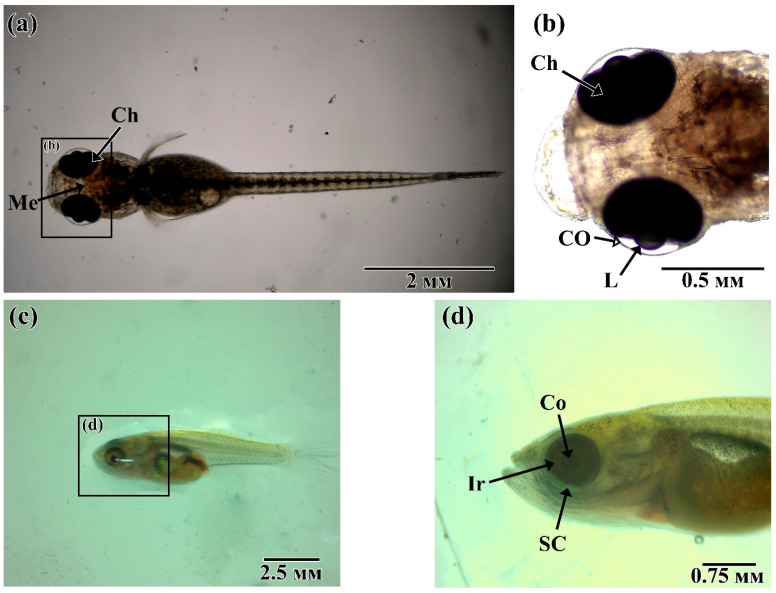
Postembryonic development of the visual system of *N. guentheri*: (**a**,**b**) 5 to 10 hph larva; (**c**,**d**) 7 dph larva. Ch—choroid; Me—melanophores; L—lens; Co—cornea, Ir—iris.

**Figure 4 animals-11-02755-f004:**
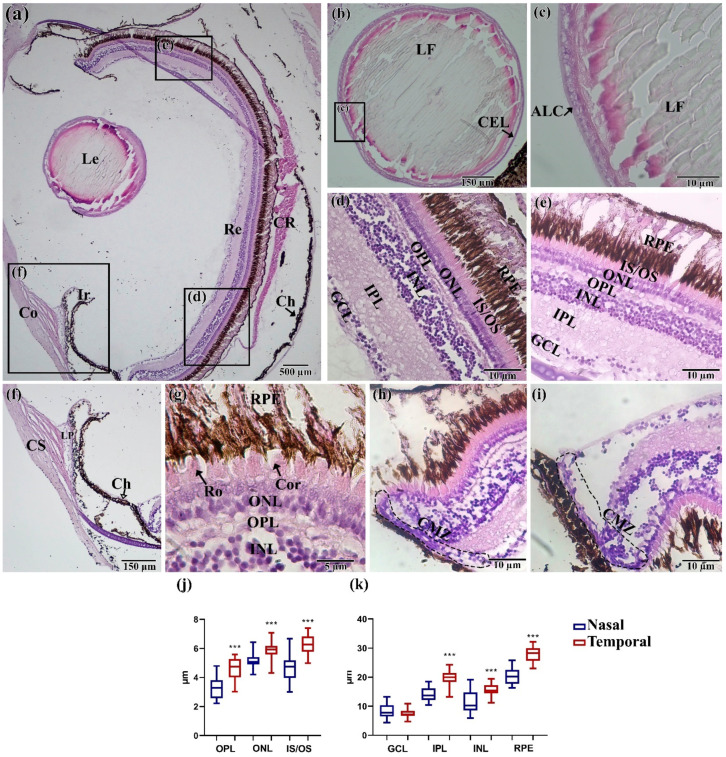
Histological structure of the eye of *N. guentheri*: (**a**) Total cut of the eye; (**b**,**c**) lens; (**d**) temporal part of the retina; (**e**) nasal part of the retina; (**f**) cornea and iris; (**g**) layer of receptor cell; (**h**,**i**) ciliary marginal zone; (**j**,**k**) thickness of layers in the temporal and nasal areas of the retina (*n* = 3 × 50). The value (***—*p* < 0.001) from one-way ANOVA with comparison using Tukey’s post hoc analysis. Le—lens, Re—retina, Co—cornea, Ir—iris, CR—choroid rete, CS—corneal stroma, Ch—choroid, GCL—ganglion cell layer, IPL—inner plexiform layer, INL—inner nuclear layer, OPL—outer plexiform layer, ONL—outer nuclear layer, IS/OS—inner segment/outer segment of the photoreceptor cells, RPE—retinal pigment epithelium/cells, LF—lens fibers, ALC—acellular lens capsule, CEL—cuboidal epithelial lens cells; LP—ligamentum pectinatum; Ro—rods; Con—cones; CMZ—ciliary marginal zone.

**Figure 5 animals-11-02755-f005:**
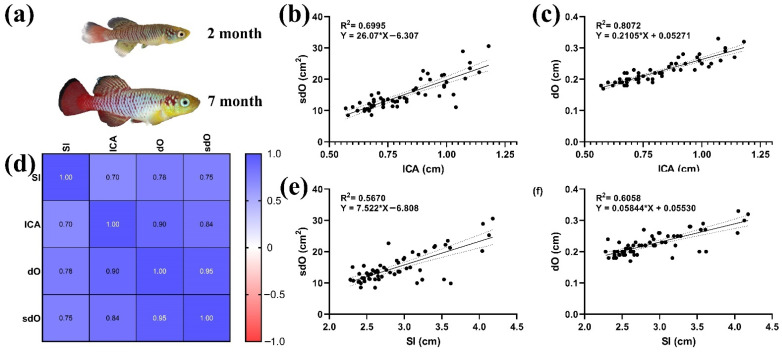
Morphoanatomical parameters of *N. guentheri* adults: (**a**) *N. guentheri* of different ages, two and seven months; (**b**) linear regression ICA and sdO; (**c**) linear regression ICA and dO; (**d**) correlation matrix of morphometric parameters; (**e**) SI and sdO linear regression; (**f**) SI and dO linear regression.

**Figure 6 animals-11-02755-f006:**
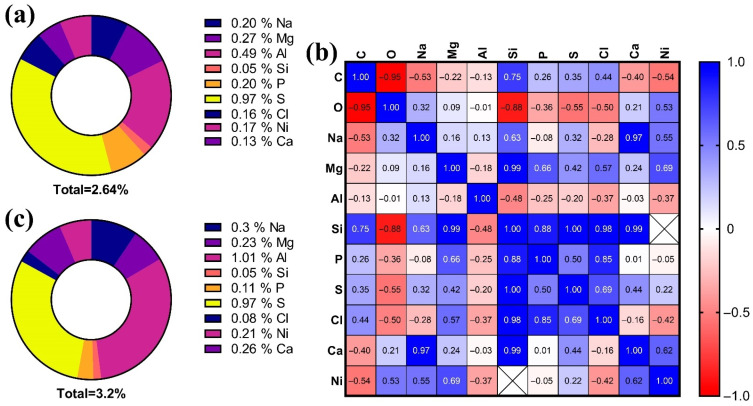
Elemental composition of the lens of *N. guentheri*: (**a**,**b**) pie diagrams of the elemental composition of the lens, excluding carbon and oxygen, for individuals two and seven months old, respectively; (**c**) correlation matrix between different elements.

**Table 1 animals-11-02755-t001:** Elemental composition (weight percent) of the lens of male *N. guentheri* aged two and seven months.

		C	O	Na	Mg	Al	Si	P	S	Cl	Ca	Ni
2 mo.	Mean	64.77	32.58	0.2	0.27	0.49	0.05	0.2	0.97	0.16	0.13	0.17
	SD	2.81	3.56	0.02	0.08	0.36	0.005	0.06	0.71	0.08	0.05	0.02
	Min	62.23	28.61	0.18	0.19	0.16	0.05	0.12	0.56	0.1	0.09	0.15
	Max	67.57	35.88	0.23	0.39	0.81	0.06	0.27	2.04	0.28	0.22	0.19
	CoF (%)	4.34	10.95	12.99	32.60	73.55	10.83	33.91	73.76	49.84	42.98	11.76
7 mo.	Mean	64.09	32.74	0.30	0.22	1.01	0.05	0.105	0.96	0.07	0.26	0.2
	SD	3.55	3.079	0.15	0.16	0.7	0	0.03	0.229	0.01	0.22	0.09
	Min	60.84	28.56	0.15	0.05	0.33	0.05	0.07	0.69	0.06	0.09	0.08
	Max	68.11	35.16	0.52	0.37	1.75	0.05	0.13	1.23	0.1	0.58	0.29
	CoF (%)	5.54	9.4	52.57	71.73	69.37	0.000	28.57	23.67	22.04	85.02	43.88

## Data Availability

The data presented in this study are openly available in FigShare at doi:10.6084/m9.figshare.16649206.

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
