# Peer review of "Some Aspects of Development and Histological Structure of the Visual System of Nothobranchius Guentheri"

_animals, 2021, doi:10.3390/ani11092755_

Round 1

Reviewer 1 Report

Comments on: “Development and histological structure of the visual system of N. guentheri”, submitted to ANIMALS. Second revision.

  • I do not understand why the authors, in the new submitted version, on line 100 added: “…occurs at 10-12 months (dph)…” dph means days post-hatching.

  • Line 202: “…It can be assumed that the differentiation of retina cells from undifferentiated neuroepithelial cells occurs at this stage…” I suggest to delete this sentence. No histological image is included in the paper to demonstrate this fact.

  • Line 227: “…the larva is acquires…”, please, correct.

  • Line 238: “…At the same time, the layer of receptor cells is represented by sensors of both types, which makes it possible to assert the high quality of vision of this fish species…” The authors included magnification (Fig. 4g) of the ONL. The morphology of rods and cones is not clearly distinguished. I suggest to include: “…at the same time, the outer nuclear layer (ONL) presents differentiated photoreceptors with well-developed outer segments…”.

  • Line 253: “…a structure like ciliary marginal zone containing undifferentiated retina cells has been isolated…” I suggest: “…”The ciliary marginal zone (CMZ) is observed in the most-peripheral retina…” The undifferentiated retinal cells are not distinguishable.

  • In panels 4 d,e, please, write IPL instead ILP (it has not been changed in the revision version of the text).

Discussion

  • Line 215: “…the emergence of new cells to the proliferation of multipotent progenitor cells located in the CMZ…” I suggest: “…the emergence of new cells in the fish retina occurs from the proliferation of multipotent progenitor cells located in the CMZ (40,41) and from dividing Müller glial cells (41)…”

Author Response

Dear reviewer, we agree with your comments. All of them were considered. Changes in the document are marked with red text and marked up using the “Track Changes” (MS Word).
Below are the answers to your specific comments.

-   I do not understand why the authors, in the new submitted version, on line 100 added: “…occurs at 10-12 months (dph)…” dph means days post-hatching.
Wording corrected for post-hatching (line 100).

-   Line 202: “…It can be assumed that the differentiation of retina cells from undifferentiated neuroepithelial cells occurs at this stage…” I suggest to delete this sentence. No histological image is included in the paper to demonstrate this fact.
Sentence deleted.

-   Line 227: “…the larva is acquires…”, please, correct.
Corrected (line 226).

-   Line 238: “…At the same time, the layer of receptor cells is represented by sensors of both types, which makes it possible to assert the high quality of vision of this fish species…” The authors included magnification (Fig. 4g) of the ONL. The morphology of rods and cones is not clearly distinguished. I suggest to include: “…at the same time, the outer nuclear layer (ONL) presents differentiated photoreceptors with well-developed outer segments…”.
Thank you for paying attention to this. Corrected (line 245).

-   Line 253: “…a structure like ciliary marginal zone containing undifferentiated retina cells has been isolated…” I suggest: “…”The ciliary marginal zone (CMZ) is observed in the most-peripheral retina…” The undifferentiated retinal cells are not distinguishable.
Sentence сorrected (line 252).

-   In panels 4 d,e, please, write IPL instead ILP (it has not been changed in the revision version of the text).
Сorrected (fig. 4).

**Discussion**

-   Line 215: “…the emergence of new cells to the proliferation of multipotent progenitor cells located in the CMZ…” I suggest: “…the emergence of new cells in the fish retina occurs from the proliferation of multipotent progenitor cells located in the CMZ (40,41) and from dividing Müller glial cells (41)…”
Sentence сorrected, according to your comments (line 315-317).

Reviewer 2 Report

The Authors of the enclosed manuscript entitled "Some aspects of development and histology of the visual system of Nothobranchius guentheri" have demonstrated the results of a very interesting study, which was focused mainly on the biological aspects of a fish species, but with a possible extrapolation to model research of visual system diseases.

Usually, such work deserves high praise, because alternatives to popular animal model species are constantly being pursued worldwide. In this case, however, it is widely known that Nothobranchius killifish are already used in quite a lot of model research, although it is another species, N. furzeri, which dominates this field of study due to its shortest average lifespan. Thus, conducting research on another species from the same genus is not necessarily groundbreaking science. I admit, though, that it deserves to be acknowledged purely from an ichthyological point of view.

Below, I arranged my specific commentary in a paragraph by paragraph manner.

Simple Summary: Very well written - this is how everyone should prepare this paragraph (many people struggle to distinguish it from the Abstract).

  • Please be consistent with the use of one of the past tenses (eg. Lines 20-22).
  • Line 19: Delete "fish".

Abstract: Nicely written, no major objections here.

  • Line 26: Delete "based on the results of our own research".
  • Line 27: Change was to "were".
  • Line 31: Change "for" to "to".

Keywords: It is commonly suggested to avoid words which already appear in the title, thus I would advise to replace "Nothobranchius guentheri", "visual system" and "histology" with some other terms, like "disease model" or something else.

Introduction: Once again, I have no major complaints. Clear and comprehensive.

  • Line 74: Change "is involved in" to "contributes to".
  • Line 88: Correct "substation" to "substantial".
  • Line 98: Delete "fish".
  • Line 100: Change "(dph)" to "post-hatching".

Material and Methods: I have spotted some minor mistakes. Apart from these, the methodological description is concise.

  • Line 105: Delete "was".
  • Line 119: Italicize "N. guentheri".
  • Line 134: Change both "was" to "were".
  • Line 134: MS-222 at 25 mg/L is as very small dose used for anaesthesia of fish, but for euthanasia, min. 250 mg/L is recommended in institutionally-approved protocols. Please explain or correct this issue.
  • Lines 136-137: It is unclear which sections were obtained, both the horizontal and frontal sections? Please indicate it more precisely.
  • Lines 146-148: How many measurements were performed in each section? What exactly was measured? Please be more precise.
  • Line 153: This time, it is indicated that sedation was performed using just 0.1 mg/L of MS-222, which is an extremely low concentration and unlike to cause any effects. Please double-check these indicated concentrations.
  • Line 170: Add "in" before "measured".

Results: I spotted some minor issues, mostly with the Figures and their descriptions, but other than that this section clearly indicates what it was supposed to show, especially the written part about the embryonic development.

  • Figure 2.: If possible, please replace the (e) and (f) subpictures with higher-magnification ones, like in subpictures (a)-(d). Furthermore, the explanation of all abbreviations was not included in the description below.
  • Line 227: Delete "is".
  • Line 237: So does the lens have any "pronounced" features, or not? The duplicated use of this word contradicts itself.
  • Line 238: Delete the comma after "stroma".
  • Line 239: Change "consisting" to "consists".
  • Figure 4.: Unfortunately, the abbreviation "Co" is repeated both for the cornea and cones. Please correct that. Moreover, the two graphs were not described fully - there should always be an indication of the applied statistical analysis, p-value and n of measurements/group.
  • Line 279: Italicize "N. guentheri".

Discussion: This paragraph was arranged in subsections in accordance with the Results and every observation mentioned in that previous part has been discussed here using proper references. Good work.

  • Line 306: Delete "as".
  • Line 310: Move the comma so that it is "Due to the shortened life cycle, by 7 dpf the larva ...".
  • Line 315: Add "occurs" after "cells".
  • Line 386: Start a new sentence after "body size".

Conclusions: Longer than in most papers in Animals, but it was justified in this case.

  • Line 437: Delete "adults".
  • Line 446: Italicize "N. guentheri".

Author Response

Dear reviewer, we agree with your comments. All of them were considered. This has significantly improved the quality of the manuscript. Changes in the document are marked with red text and marked up using the “Track Changes” (MS Word).
Below are the answers to your specific comments.

**Simple Summary:** 

-   Please be consistent with the use of one of the past tenses (eg. Lines 20-22).
Sentence сorrected.

-   Line 19: Delete "fish".
Сorrected.

**Abstract:** 

-   Line 26: Delete "based on the results of our own research".
-   Line 27: Change was to "were".
-   Line 31: Change "for" to "to".
Сorrected.

**Keywords:** 
Keywords changed.

**Introduction:** 

-   Line 74: Change "is involved in" to "contributes to".
-   Line 88: Correct "substation" to "substantial".
-   Line 98: Delete "fish".
-   Line 100: Change "(dph)" to "post-hatching".
Comments corrected.

**Material and Methods:** 

-   Line 105: Delete "was".
-   Line 119: Italicize "N. guentheri".
-   Line 134: Change both "was" to "were".
Comments corrected.

-   Line 134: MS-222 at 25 mg/L is as very small dose used for anaesthesia of fish, but for euthanasia, min. 250 mg/L is recommended in institutionally-approved protocols. Please explain or correct this issue.
Thank you for paying attention to this. Corrected to 250 mg / L.

-   Lines 136-137: It is unclear which sections were obtained, both the horizontal and frontal sections? Please indicate it more precisely.
On the prepared total sections of the fish body, the eye was cut in the horizontal plane, according to the work Howland, H. C. et. al. Vision research, 48(18) pp. 1926-1927.

-   Lines 146-148: How many measurements were performed in each section? What exactly was measured? Please be more precise.
Clarifications added.

-   Line 153: This time, it is indicated that sedation was performed using just 0.1 mg/L of MS-222, which is an extremely low concentration and unlike to cause any effects. Please double-check these indicated concentrations.
The dose was selected empirically to reduce physical activity when photographing fish.

-   Line 170: Add "in" before "measured".
Сorrected.

**Results:** 

-   Figure 2.: If possible, please replace the (e) and (f) subpictures with higher-magnification ones, like in subpictures (a)-(d). Furthermore, the explanation of all abbreviations was not included in the description below.
Higher-magnification subpictures added. Abbreviations added.

-   Line 227: Delete "is".
-   Line 237: So does the lens have any "pronounced" features, or not? The duplicated use of this word contradicts itself.
-   Line 238: Delete the comma after "stroma".
-   Line 239: Change "consisting" to "consists".
Сorrected (line 236,238).

-   Figure 4.: Unfortunately, the abbreviation "Co" is repeated both for the cornea and cones. Please correct that. Moreover, the two graphs were not described fully - there should always be an indication of the applied statistical analysis, _p_-value and _n_ of measurements/group.
Сorrected (line 257, 261).

-   Line 279: Italicize "N. guentheri".
Сorrected.

**Discussion:** 

-   Line 306: Delete "as".
-   Line 310: Move the comma so that it is "Due to the shortened life cycle, by 7 dpf the larva ...".
-   Line 315: Add "occurs" after "cells".
-   Line 386: Start a new sentence after "body size".
Comments corrected (line 316, 386).

**Conclusions:** 

-   Line 437: Delete "adults".
-   Line 446: Italicize "N. guentheri".
Comments corrected.

This manuscript is a resubmission of an earlier submission. The following is a list of the peer review reports and author responses from that submission.

Round 1

Reviewer 1 Report

Comments on: “Development and histological structure of the visual system of N. guentheri”, submitted to ANIMALS.

The development of the visual system of N. guentheri (a non standard-model organism) has been studied in the embryonic and posthatching periods. The present research shows that the main eye structures were formed before the hatching stage. Authors show interesting data about lens aging. However, some aspects about retinal development need to be revised. In some cases, “Results” section is speculative. Major revisions need to be included in the revised version of the study.

A previous study on adult retinal neurogenesis in N. furzeri has been published (Tozzini et al., 2012 Aging Cell: 11, pp241–251). This study demonstrates the existence of a neuroepithelial ciliary marginal zone, indicating the continuous growth of the retina in this fish species throughout animal´s life. It could be an important article to be included in the “Discussion” section, when the authors discuss the growth of the eye structures.

RESULTS

  • Line 185: “…begins with the formation of the 185 optic (eye) vesicle at 16-18 hours post-fertilization (hpf)…” I suggest to include “(not shown)”
  • Line 187: “…optic vesicle ends at 24-26 hpf (Figure 2 a)”… Please, include Figure 2a,b.

Why do you know that the formation of the optic vesicle ends by this stage? Histological sections could help to demonstrate that the optic vesicle/optic cup is already formed.

  • Line 195: “…an active division of the periocular… “ the authors did not detect the active division (i.e. mitotic activity by using antibodies or detecting mitoses with histological stanining). I suggest to delete this sentence.
  • Line 196: “…It can be assumed that, at this stage, differentiation of retinal cells from the retinal pigment epithelium occurs…” I do not understand this sentence. Retinal cell differentiation occurs from retinal stem cells: undifferentiated neuroepithelial cells in the neuroblastic layer of the retina, differentiating Müller glia, or neuroepithelial cells located in the ciliary marginal zone. Retinal neurons in the vertebrate retina differentiated from the retinal pigment epithelium under certain regenerative stimulus in amphibians and chicken embryo, but not in fish. I suggest to delete this sentence.

  • Figure 2d is not described in the text.

  • Line 210: “…In the eye of the embryo, the pupil and the iris are clearly distinguishable, and the eye actively reacts to changes in illumination (Figure 2f)…” I suggest: “…In the eye of the embryo, the pupil and the iris are clearly distinguishable (Figure 2 f), and the eye actively reacts to changes in illumination (unpublished data?-not shown?).

  • Line 221: “…At 7-10 days post hatching (dph), the larva is fully developed and its visual system functions as in adults…” how do you know?. In some cases, pigmentation in the RPE is clearly seen but the retina still shows some features of immaturity (i.e. poorly developed photoreceptors). Maybe by these early posthatching stages the larva responds to movement but visual capabilities are not fully developed. Histological sections of the retina by these stages could demonstrate the maturational stage of the retina. It is very important to know if the retina is fully developed at the hatching stage, because visual capabilities at hatching are determined by the structure of the neural retina (see Evans and Browman, 2004; American Fisheries Society Symposium 40:145–166; Bejarano-Escobar et al., 2014; Rev Fish Biol Fisheries 24:127–158; Álvarez-Hernán et al., 2019; Zoomorphology 138:371–385).

  • Figure 3 is not described in the text.
  • Line 238: “…At the same time, the layer of receptor cells is represented by sensors of both types, which makes it possible to assert the high quality of vision of this fish species…” The authors did not show the presence of rods and cones. A new panel showing magnification of the ONL in (Fig. 4 d, e) could help to demonstrate this sentence.

  • Line 243: “…retina of the eye has its own choroid rete, similar to Danio rerio…” Please, include the reference.

  • Figure 4: I suggest to include magnifications of the ciliary marginal zone showing undifferentiated cells that could suggest the continuous growth of the retina (according to Tozzini et al., 2012).

In panels 4 d,e, please, write IPL instead ILP.

Discussion

  • Line 299: “…this work shows that the beginning of the formation of the visual system occurs at 16-18 hpf and ends at 21-24 dpf…”I suggest to delete that visual system differentiation ends at 21-24 dpf. It is not clearly demonstrated with the data included in the study.
  • Line 303: “…Due to the life cycle being shortened by 7 dph, the fry can be considered fully formed and the main morphoanatomical characteristics correspond to adult fish…” I suggest to delete this sentence. 7dph larva (Figure 3) does not show the morphoanatomical characteristics of the juvenile or adult specimens.
  • Line 325: “…At the time of hatching, the visual system is fully formed. After 7 dph, the fry acquires the features that are characteristic of adults, and the development of the visual system can be considered complete…”Again, authors do not show enough information to demonstrate that visual system is fully formed at hatching (see above). A simple histological analysis of the retina in recent hatched animals could give abundant information about the maturity of the visual system at hatching.

  • Line 330: “…As noted by Seritrakul et al. [2], the structure of the retina is very similar in all vertebrates, including humans, which makes the study of the visual system of bony fish is relevant from the point of view of biomedicine…”. I suggest to include in this sentence that postnatal neurogenesis and the regenerative capacity of the fish retina (due to the presence of stem cells) are the main features that make the fish retina an excellent model in biomedicine.

Reviewer 2 Report

The article “Development and histological structure of the system of Bothobranchius guentheri” is an introduction to some aspects of the eye of an annual killifsh. The work is a miscellany of disparate quality in the different sections.

The model is of undoubted scientific interest. It merits detailed studies of each of the sections covered in this paper. In this sense, the article could be considered as a first approximation to general aspects of the eye of this annual killifish.

- The article highlights the good quality of the micrographs.

- The introduction is clear and pertinent.

- As for the longevity of the animals, the period indicated on line 100 is not clear whether it is dph or dpf, which can be very significant outside the laboratory.

- In "object of study" the number of animals used in each experiment is of greater interest than the number of initial breeders.

- Line 112 does not specify the units of measurement (cm).

- Is the reference on line 129 confused?

- Why 7 month old animals? They are adult breeders, but animals from different post-hatching periods are necessary to analyses the histological evolution in growth, maturity and ageing.

- The use of a single general morphological technique limits the study considerably.

- It is essential to specify the section plane (using the usual terms). The terms caudal and cranial are not suitable for unequivocal histological orientation.

- Lines 196-197 “It can be assumed that, at this stage, differentiation of retinal cells fron the retinal pigment epithelium occurs”. This sentence is badly worded, because in its present form it expresses a totally wrong concept.

- The contents of lines 221-220 are not experimental results of this work. They should therefore be associated with the appropriate bibliography.

- The description of the retinal layers is poor and incomplete (no optic nerve fibre layer?).

- On the other hand, accepting adequate differentiation between cones and rods with haematoxylin eosin, density claims require statistical study.

- The layer thickness comparison requires specifying criteria for zone choice and section selection.

- An unaddressed aspect closely related to the development, growth and ageing of the retina is the peripheral germinal zone. This area determines the growth of the retina by the addition of new cells. The possible variations in this area determine how retinal growth evolves (or not). This supports the need to use animals of different ages.

- Morphoanatomical measurements are adequate in method and results are consistent.

- The study of the composition of the lens is, in my opinion, the most interesting part of the study.

- In the conclusions section, what is stated in paragraphs 2 and 3 (lines 413-417; 418-420) are indirect assessments, only partially based on the results of the work.

- The bibliography is current and adequate. However, it will be helpful if all references have the same criteria (28).

- Reference 66 (line 139) is incorrect (26?).

- In line 312 the reference 39 is not appropriate.

- In the text, the numbering of the references is not in the right order on lines 294, 382 and 383.

References 11 and 12 are the same.